# Diabetes Mellitus screening and associated factors in Peru: A cross-sectional analysis of a national health survey

Einer Carlos Eduardo Arevalo-Rios[1]*, Jennifer Layza-Reyes[1],
Victor Hugo Noriega-Ruiz[1,2]

1 Alberto Hurtado School of Medicine, Universidad Peruana Cayetano Heredia, Lima, Peru, 2 Department of Endocrinology, Clinica Anglo Americana, Lima, Peru

* einer.arevalo@upch.pe

## Abstract

Screening rates for Type 2 diabetes mellitus (T2DM) in Peru remain low despite an increasing prevalence of the disease. In 2022, the American Diabetes Association (ADA) recommended screening adults aged 18–34 with overweight or obesity and at least one risk factor, including Latino ethnicity and lowered the universal screening age to 35 years. This study aimed to determine the T2DM screening prevalence in Peru and the factors associated with lack of screening in the population that meets ADA screening criteria. A cross-sectional, secondary data analysis was conducted based on the 2022 Peruvian Demographic and Health Survey database. Screening was defined as apparently healthy individuals having their blood glucose measured in the last year. Participants aged 18–34 years with overweight or obesity, as well as all participants aged ≥ 35 years, were included. Bivariate analysis was performed using the chi-squared test, and multivariate Poisson regression was used to estimate prevalence ratios (PRs), adjusting for potential confounders. Of 26,166 individuals who met inclusion criteria, 25.3% were screened for T2DM. The factors most strongly associated with lack of screening were: age 18–34 years (aPR: 1.08; 95% CI: 1.05-1.11), having only elementary education (aPR: 1.18; 95% CI: 1.13-1.23), not having health insurance (aPR: 1.15; 95% CI: 1.12-1.18), belonging to the lowest wealth quintile (aPR: 1.09; 95% CI: 1.05-1.13), and daily smoking (aPR: 1.10; 95% CI: 1.01-1.19). The national rate of screening for T2DM in Peru is low (25.3%). There is a higher prevalence for a lack of screening in people who: are younger, have only primary education, do not have health insurance, belong to the lowest wealth quintile, and are smokers. Targeted interventions are needed to improve screening coverage in these high-risk populations.

**Data availability statement:** All relevant data are within the paper and its Supporting Information files.

**Funding:** The authors received no specific funding for this work.

**Competing interests:** The authors have declared that no competing interests exist.

## Introduction

Diabetes Mellitus is one of the leading causes of mortality worldwide. However, early diagnosis and treatment can significantly reduce morbidity and mortality [1]. Longer periods without adequate metabolic control increase the risk of developing chronic complications that worsen prognosis [1,2]. Screening with a fasting plasma glucose measurement, glycated hemoglobin (HbA1c), or glucose tolerance test is the main strategy for early detection of this disease. In 2022, the ADA modified its screening recommendations to include universal screening for all individuals starting at age 35; and targeted screening for individuals over 18 years old with overweight or obesity and at least one additional risk factor, including Latino ethnicity, which is considered high-risk [3]. These modifications were made to improve sensitivity of screening and to promote early detection of disease. Following these recommendations, and considering Peruvians as Latinos, all individuals with overweight or obesity between the ages of 18 and 34, as well as everyone over 35, should be screened for T2DM in Peru.

Peru is considered an upper-middle-income country with a geographically and ethnically diverse population that has persistent barriers to healthcare access despite recent economic growth [4]. Its diverse topography is marked by differences in the three main natural regions of the country: coast, highlands and jungle. Approximately 58% of the population resides on the coast, with many of the healthcare facilities concentrated in the capital districts of each department. The highlands of Peru are inhabited by 28.1% of the population, with many of them living at high altitudes, with limited infrastructures, deficient road networks and poor access to health services. The jungle of Peru, comprising 13% of the population, has many scattered native-indigenous communities with limited interconnection and many of them only accessible by air or water [4]. These characteristics pose an intrinsic barrier to healthcare access throughout the nation that impact early disease detection, including T2DM screening. Regarding non-communicable diseases, the prevalence of T2DM in Peru has increased in the past decades, with latest reports estimating a prevalence of 5–7% highlighting variations among regions in the nation (coast, highlands, and jungle) [5–7]. Interestingly, previous studies have reported lower diabetes prevalence at higher altitude in Peru [8]. It is hypothesized that socioeconomic and cultural barriers affect the T2DM screening coverage. It is currently unknown how many people have been screened in the country following the latest ADA guideline recommendations. While the Ministry of Health in Peru has defined screening criteria in national guidelines, these have not been updated since 2016, despite the increasing prevalence of the disease in the country [5,7]. Since T2DM is a disease that leads to complications that can be prevented with timely diagnosis and treatment, this study uses the latest and more sensitive screening criteria, which are the ADA screening recommendations. Recent data from a nationally representative health survey was analyzed to find the prevalence and risk factors associated with lack of diabetes screening in Peru.

## Materials and methods

### Study design and area

A cross-sectional analysis was performed using the 2022 Peruvian Demographic and Health Survey (DHS) database. This survey was conducted by the National Institute of Statistics and Informatics (INEI) from January to December 2022 on a nationally representative sample in all regions and departments of Peru. The DHS collected information through face-to-face interviews, and anthropometric measurements were obtained by trained individuals. The survey comprises multiple questionnaires; for this study, the Health Survey component was analyzed, as it includes data on non-communicable diseases such as T2DM. These datasets are publicly accessible at https://proyectos.inei.gob.pe/microdatos/ (S1–S2 Data).

### Study population

The target population of the DHS are the usual residents of the interviewed households, which provide representative estimates of the population of Peru. In this study, responses from those who met the 2022 ADA screening criteria were analyzed, considering Latino/Hispanic ethnicity as a risk factor. Body Mass Indexes were calculated based on the anthropometric measures obtained in the interviews and as recorded in the Health Survey.

### Inclusion and exclusion criteria

Participants were included based on the two key populations to screen according to the ADA:

Adults aged 18–34 years with overweight or obesity and one additional risk factor, specifically Latino ethnicity; and all individuals aged 35 years and older. Only participants with complete data for the Health Questionnaire were included.

Pregnant women, individuals with a previous diagnosis of T2DM and those taking antidiabetic medications were excluded.

### Data collection and sample size

The DHS used a two-stage, probabilistic, balanced, stratified, and independent sampling design, covering all regions of Peru. The Health Survey was administered to 36 650 households and interviewed 31 917 individuals aged 15 years and over. This study involved no direct contact with participants, as it was based on the analysis of publicly available datasets. Only complete-case responses were analyzed.

### Variables

The dependent variable was the self-report of T2DM screening; based on the question: "In the past 12 months, has a doctor or other health professional measured your blood glucose or "sugar" level?". This outcome was categorized as a dichotomous variable.

The following independent variables were included in the analysis. Sociodemographic variables included age (18–34 and 35 or more), gender (male and female), area of residence (rural and urban), region of origin (Lima, other coastal region, highlands, and jungle), altitude of residence (less than 3000 m.a.s.l. and more than 3000 m.a.s.l.), educational level, ethnicity (native, afrodescendant, mestizo, white, and other), and wealth index. Anthropometric measurements, weight (in kg) and height (in cm), were used to calculate Body Mass Index (BMI). Clinical variables included previous hypertension diagnosis, alcohol consumption, insurance status, and smoking status. These variables were selected based on potential risk factors for the lack of screening from literature review. Further description of variables is provided as supplementary information (S1 Table).

## Statistical analysis

Data was analyzed using STATA SE (18.0 version, College Station, Texas 77845, USA). To account for the complex sampling design of the DHS and the analysis of a subpopulation, the *svy* and *subpop* commands were used. Raw databases were merged to include the sociodemographic, anthropometric and health questionnaire information of individuals. Categorical variables are reported as frequencies and proportions, while quantitative variables are reported as mean and standard error for normally distributed data. Chi-square tests were used for the bivariate analysis between the independent variables and the dependent variable. Multivariate analysis was conducted using generalized linear models with Robust Poisson regression variance estimation to calculate crude and adjusted prevalence ratios for non-screening, considering the expected outcome prevalence to be greater than 10% [9]. The variables included in the multivariate model were the ones that showed association in the bivariate model and those epidemiologically relevant according to the literature review. A p-value ≤ 0.05 was considered statistically significant.

## Ethical considerations

Data from the DHS is available publicly online as published by the INEI. When conducting the field interviews, the interviewers obtained written informed consent from each respondent before participating in the survey. For this study, data was accessed on June 26th, 2024. The information collected did not allow for the identification of individuals. As this is a secondary study and no direct contact with participants was made, informed consent was not required. The protocol of this study was reviewed and approved by the Institutional Committee on Ethics in Human Research of the Peruvian University Cayetano Heredia in Lima, Peru.

## Results

### Characteristics of the study population

The subpopulation that met the ADA screening criteria (35 years or older and 18–34 years with overweight or obesity) was 26,166 individuals (Fig 1). Among them, the majority were women (52.5%), over 35 years old, resided in urban areas, identified as "Mestizo" ethnicity, and had health insurance. Approximately one-third had achieved higher education. Participants were evenly distributed across the five wealth quintiles, which were calculated based on household living standards. Most participants lived in Metropolitan Lima. The majority did not smoke, 11.4% had been diagnosed with hypertension, and approximately one-third of individuals reported alcohol consumption in the past 30 days. The full description of the characteristics is shown in Table 1.

### Prevalence of T2DM screening and associated factors

Among individuals who met the ADA T2DM screening criteria, 25.3% had a blood sugar measurement in the last 12 months. Bivariate analysis (Table 2) revealed that screening was less common among the rural population (85.7%), individuals without education (80.5%) and those from lower socioeconomic levels. Screening prevalence was lower in the highlands (80.9%), especially at an altitude exceeding 3,000 meters above sea level (83.1%). Additionally, individuals without health insurance (84.6%), those without a diagnosis of hypertension (77.3%), and daily smokers (83.1%) were less likely to be screened. Alcohol consumption was not associated with screening prevalence (p = 0.7596).

Multivariate analysis identified several factors associated with the lack of screening (Fig 2). These were: age between 18–34 years (aPR: 1.08; 95%CI: 1.05-1.11), male sex (aPR: 1.00063; 95%CI: 1.00063-1.5), belonging to the lowest wealth quintile (aPR: 1.09; 95%CI: 1.05-1.13), having only elementary education (aPR: 1.18; 95%CI:

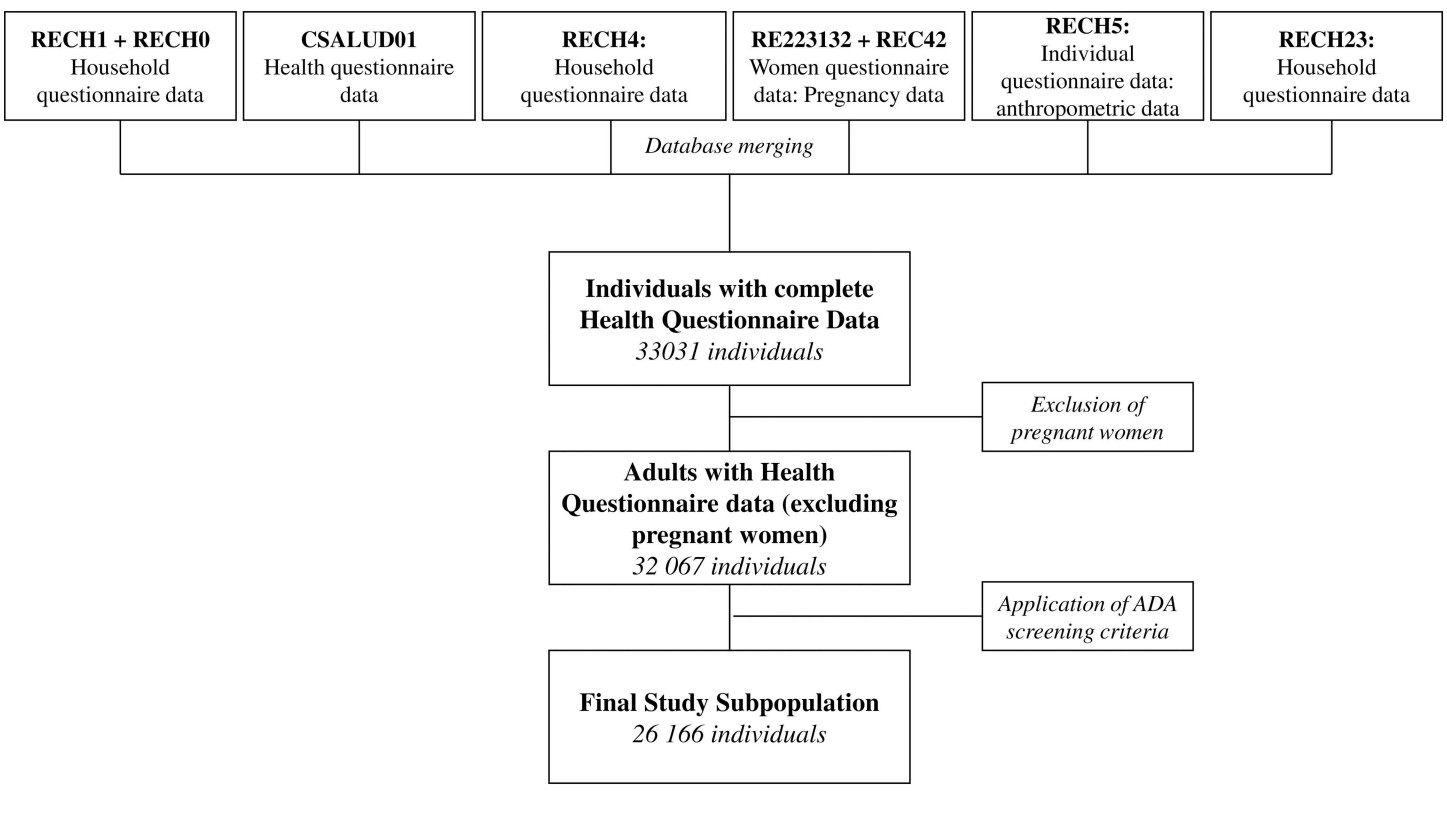

**Fig 1. Flowchart of participant selection.**

1.13-1.23), living at an altitude greater than 3,000 m.a.s.l (aPR: 1.04; 95%CI: 1.004-1.07), not having health insurance (aPR: 1.15; 95%CI: 1.12-1.18), and being a daily smoker (aPR: 1.10; 95%CI: 1.01-1.19). Factors that increased the prevalence of screening included having hypertension (aPR: 0.73; 95%CI: 0.68-1.78) and being in the two highest wealth quintiles (fourth quintile aPR: 0.93; 95%CI: 0.90-0.97; fifth quintile aPR: 0.85; 95%CI: 0.81-0.89). No significant association was found for ethnicity, area of residence, or region of origin. Further details can be found in Table 3.

## Discussion

In this nationally representative analysis, only 25.3% of eligible individuals in Peru were screened for T2DM based on the 2022 ADA criteria. This prevalence is lower than estimates from previous national studies, such as 27.7% in 2020 and 30.6% in 2014 [10,11]. However, these studies did not apply updated ADA criteria and included broader age groups, which may explain these different estimates. This is the first study in Peru to apply the revised ADA recommendations, which prioritize early detection.

Compared to other Latin American countries, screening rates in Peru are limited. In Mexico, 15.2% of adults accessed preventive diabetes services, while in Brazil, over 70% underwent glucose testing within the previous two years [12,13]. These disparities likely reflect differences in national screening protocols, primary care access, and methodological approaches.

**Table 1. Characteristics of the population that met the ADA screening criteria.**

| Characteristics | Total (N = 26 166) | Weighted proportion % (95 CI) |
|---|---|---|
| **Age (mean, SE)** | 45.39 (0.18) | – |
| **Age (categorized)** | | |
| 18–34 years | 7 251 | 27.7 (26.8 – 28.6) |
| 35 or more years | 18 915 | 72.3 (71.4 – 73.2) |
| **Sex** | | |
| Male | 12 440 | 47.5 (46.4 – 48.6) |
| Female | 13 726 | 52.5 (51.4 – 53.6) |
| **Area of residence** | | |
| Rural | 4 830 | 18.5 (17.9 – 19.0) |
| Urban | 21 336 | 81.5 (81.0 – 82.1) |
| **Education level** | | |
| No education | 1 045 | 4.0 (3.6 – 4.4) |
| Incomplete elementary | 4 126 | 15.8 (15.0 – 16.6) |
| Complete elementary | 1 473 | 5.6 (5.2 – 6.0) |
| Incomplete secondary | 3 332 | 12.7 (12.0 – 13.5) |
| Complete secondary | 7 456 | 28.5 (27.5 – 29.5) |
| Higher education | 8 734 | 33.4 (32.3 – 34.5) |
| **Ethnicity** | | |
| Native | 7 335 | 28.0 (27.1 – 29.0) |
| Zambo, afro descendant | 2 998 | 11.5 (10.8 – 12.2) |
| White | 1 842 | 7.0 (6.5 – 7.6) |
| Mestizo | 12 349 | 47.2 (46.1 – 48.3) |
| Other | 327 | 1.3 (1.0 – 1.5) |
| Does not know | 1 315 | 5.0 (4.6 – 5.5) |
| **Economic level** | | |
| The poorest | 4 839 | 18.5 (17.8 – 19.2) |
| Poor | 5 216 | 19.9 (19.1 – 20.8) |
| Middle | 5 555 | 21.2 (20.3 – 22.2) |
| Rich | 5 422 | 20.7 (19.7 – 21.8) |
| The richest | 5 133 | 19.6 (18.5 – 20.8) |
| **Region of origin** | | |
| Metropolitan Lima | 9 789 | 37.4 (36.3 – 38.6) |
| Rest of the coast | 6 848 | 26.2 (25.2 – 27.2) |
| Highlands | 6 363 | 24.3 (23.3 – 25.4) |
| Jungle | 3 166 | 12.1 (11.4 – 12.8) |
| **Altitude** | | |
| Less than 3 000 m.a.s.l. | 22 291 | 85.2 (84.3 – 86.1) |
| Great or equal to 3 000 m.a.s.l. | 3 876 | 14,8 (13.9 – 15.7) |
| **Health insurance** | | |
| Yes | 21 539 | 82.3 (81.4 – 83.2) |
| No | 4 627 | 17.7 (16.8 – 18.6) |
| **Hypertension** | | |
| Present | 2 982 | 11.4 (10.7 – 12.1) |
| Absent | 23 184 | 88.6 (87.8 – 89.2) |
| **Daily smoker** | | |
| No | 25 780 | 98.5 (98.2 – 98.8) |

*(Continued)*

**Table 1.** (Continued)

| Characteristics | Total (N = 26 166) | Weighted proportion % (95 CI) |
|---|---|---|
| Yes | 386 | 1.5 (1.2 – 1.8) |
| **Alcohol drinking** | | |
| Present | 16 182 | 61.8 (60.7 – 62.9) |
| Absent | 9 984 | 38.2 (37.1 – 39.3) |

SE = Standard error, m.a.s.l. = meters above sea level

Age was an associated factor for screening. Individuals aged 18–34 years were less likely to undergo screening, despite meeting ADA criteria. Previous studies report lower healthcare utilization in younger adults, who may have lower perceived susceptibility to chronic diseases [10,12–19]. This is concerning given the growing incidence of early-onset T2DM [20].

Additionally, men were less likely to be screened than women, consistent with evidence from multiple countries [10,14,21]. Differences in how men seek healthcare, work obligations, and cultural norms, may explain this gap [16,19,22]. Similarly, individuals with lower educational level and those in the lowest wealth quintiles were less likely to be screened, likely due to disparities in health literacy, financial resources, and healthcare access [10,11,15,17].

Lack of health insurance was strongly associated with lower screening rates. This factor has been described as a predictor for low healthcare utilization, and lower opportunities for screening [23–25]. In contrast, individuals with a prior diagnosis of hypertension were more likely to be screened, likely reflecting more frequent contact with healthcare providers and awareness of comorbidities [12,17,23]. Daily smoking was associated with reduced screening, aligning with previous findings that link unhealthy behaviors to lower use of preventive health services [12,17,26].

Although no significant difference was observed between urban and rural residence, individuals living at ≥3,000 meters above sea level had lower screening rates. This may reflect geographical barriers to care in high-altitude Andean communities, where transportation challenges and limited healthcare infrastructure reduce access to routine medical services [27].

These findings highlight the need for targeted public health strategies to increase screening uptake, particularly among young adults, men, individuals with lower socioeconomic status, the uninsured, and those in geographically remote areas. Expanding access to primary care and community-based screening programs could help reduce disparities and improve early detection of T2DM.

In addition to addressing these gaps, adopting the ADA screening criteria in Peru may also represent a cost-effective strategy to enhance early detection. In the United States, the number needed to screen (NNS) is 15 for individuals aged ≥35 years and 12 for those aged ≥45 years, with estimated costs per positive case of $66.37 and $55.53, respectively [28]. While broader screening increases initial costs, it substantially reduces the number of missed diagnoses. Country-specific cost-effectiveness analyses are needed to assess the feasibility and impact of implementing these recommendations in Peru.

This study has several strengths. It is based on a large, nationally representative sample drawn using a probabilistic design, enhancing generalizability. It is also the first to estimate T2DM screening coverage in Peru using the updated ADA criteria. It also provides new insights into the sociodemographic and health-related determinants of screening in a middle-income setting.

However, limitations should be acknowledged. The cross-sectional design impedes causal inference. Recall bias may be present due to the use of self-reported data. The outcome of self-reported glucose testing in the past

**Table 2. Factors associated with the lack of T2DM screening in the last 12 months: Bivariate analysis.**

| Variables | Glucose testing in the last 12 months (N = 26 166) | | |
|---|---|---|---|
| | No (n = 19546) | Yes (n = 6 620) | p* |
| | nº (%) 74.7% | nº (%) 25.3% | |
| **Age (categorized)** | | | |
| 18–34 years | 5 783 (79.8%) | 1 468 (20.2%) | <0.001 |
| 35 or more years | 13 769 (72.8%) | 5 147 (27.2%) | |
| **Sex** | | | |
| Female | 10 061 (73.3%) | 3 666 (26.7%) | 0.002 |
| Male | 9 490 (76.3%) | 2 949 (23.7%) | |
| **Area of residence** | | | |
| Rural | 4 140 (85.7%) | 688 (14.3%) | <0.001 |
| Urban | 15 409 (72.2%) | 5 927 (27.8%) | |
| **Education level** | | | |
| No education | 843 (80.5%) | 204 (19.5%) | <0.001 |
| Incomplete elementary | 3 297 (79.9%) | 830 (20.1%) | |
| Complete elementary | 1 287 (87.4%) | 186 (12.6%) | |
| Incomplete secondary | 2 601 (78.1%) | 730 (21.9%) | |
| Complete secondary | 5 730 (76.9%) | 1 724 (23.1%) | |
| Higher education | 5 793 (66.3%) | 2 938 (33.7%) | |
| **Ethnicity** | | | |
| Native | 5 694 (77.6%) | 1 643 (22.4%) | <0.001 |
| Zambo, afrodescendant | 2 360 (78.7%) | 639 (21.3%) | |
| White | 1 376 (74.8%) | 463 (25.2%) | |
| Mestizo | 8 839 (71.6%) | 3 512 (28.4%) | |
| Other | 249 (75.9%) | 79 (24.1%) | |
| Do not know/No response | 1 034 (78.7%) | 280 (21.3%) | |
| **Economic level** | | | |
| The poorest | 4 210 (87.0%) | 628 (13%) | <0.001 |
| Poor | 4 189 (80.3%) | 1 028 (19.7%) | |
| Middle | 4 228 (76.1%) | 1 327 (23.9%) | |
| Rich | 3 807 (70.2%) | 1 614 (29.8%) | |
| The richest | 3 116 (60.7%) | 2 017 (39.3%) | |
| **Region of origin** | | | |
| Metropolitan Lima | 6 863 (70.1%) | 2 925 (29.9%) | <0.001 |
| Rest of the coast | 5 050 (73.8%) | 1 798 (26.3%) | |
| Highlands | 5 147 (80.9%) | 1 217 (19.1%) | |
| Jungle | 2 488 (78.6%) | 678 (21.4%) | |
| **Altitude** | | | |
| Less than 3 000 m.a.s.l. | 16 328 (73.2%) | 5 963 (26.8%) | <0.001 |
| Great or equal to 3 000 m.a.s.l. | 3 224 (83.1%) | 654 (16.9%) | |
| **Health insurance** | | | |
| Yes | 15 637 (72.6%) | 5 903 (27.4%) | <0.001 |
| No | 3 914 (84.6%) | 712 (15.4%) | |
| **Hypertension** | | | |
| Present | 1 622 (54.4%) | 1 361 (45.6%) | <0.001 |
| Absent | 17 924 (77.3%) | 5 259 (22.7%) | |

*(Continued)*

**Table 2.** (Continued)

| Variables | Glucose testing in the last 12 months (N = 26 166) | | |
|---|---|---|---|
| | No (n = 19546) | Yes (n = 6 620) | p* |
| | nº (%) 74.7% | nº (%) 25.3% | |
| **Daily smoker** | | | |
| No | 319 (83.1%) | 65 (16.9%) | 0.0473 |
| Yes | 19 232 (74.6%) | 6 549 (25.4%) | |
| **Alcohol drinking** | | | |
| Present | 7 439 (74.5%) | 2 543 (25.5%) | 0.7596 |
| Absent | 12 110 (74.8%) | 4 071 (25.2%) | |

*p-value from chi-square test

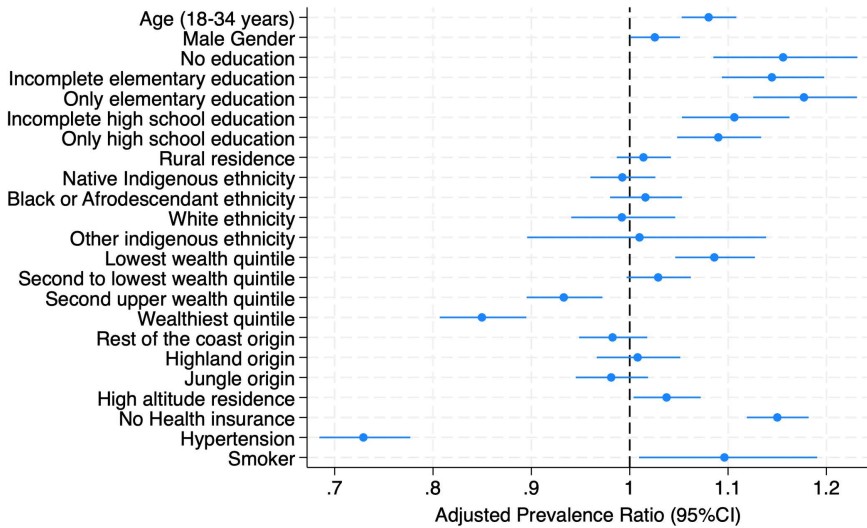

**Fig 2. Forest plot for factors associated with the lack of screening for T2DM.**

year may not always reflect screening for T2DM, as glucose may be tested in other clinical contexts. Additionally, the survey does not include information on other predictors of screening, such as family history of diabetes. Finally, the lack of biomarker data prevents assessment of undiagnosed diabetes and limits estimation of a true screening gap.

## Conclusion

Only 25.3% of the population that met the ADA diabetes screening criteria, underwent screening in Peru in 2022. The factors associated with the lack of screening included being in the age range of 18–34 years, being male, having a low level of education, belonging to the lower wealth quintiles, lacking health insurance, living at an altitude of 3,000 m.a.s.l or higher, and being a daily smoker. In this regard, efforts should be made to promote public policies that raise awareness about the importance of early diagnosis of T2DM and that target the population with these characteristics.

PLOS Global Public Health

**Table 3. Factors associated with the non-screening for T2DM: Multivariate analysis.**

| Variables | Crude model | | | Adjusted model | | |
|---|---|---|---|---|---|---|
| | PR | CI 95% | p | PR | CI 95% | p |
| **Age** | | | | | | |
| 35 or more years | 1.00 | | | 1.00 | | |
| 18–34 years | 1.10 | 1.07–1.12 | <0.001 | 1.08 | 1.05–1.12 | <0.001 |
| **Sex** | | | | | | |
| Female | 1.00 | | | 1.00 | | |
| Male | 1.04 | 1.01–1.07 | 0.002 | 1.03 | 1.00–1.05[a] | 0.044 |
| **Area of residence** | | | | | | |
| Urban | 1.00 | | | 1.00 | | |
| Rural | 1.19 | 1.16–1.21 | <0.001 | 1.01 | 0.99–1.04 | 0.320 |
| **Education level** | | | | | | |
| Higher education | 1.00 | | | 1.00 | | |
| No education | 1.21 | 1.15–1.29 | <0.001 | 1.16 | 1.09–1.23 | <0.001 |
| Incomplete elementary | 1.20 | 1.16–1.25 | <0.001 | 1.15 | 1.09–1.20 | <0.001 |
| Complete elementary | 1.32 | 1.26–1.37 | <0.001 | 1.18 | 1.13–1.23 | <0.001 |
| Incomplete secondary | 1.18 | 1.12–1.24 | <0.001 | 1.11 | 1.05–1.16 | <0.001 |
| Complete secondary | 1.16 | 1.11–1.20 | <0.001 | 1.09 | 1.05–1.13 | <0.001 |
| **Ethnicity** | | | | | | |
| Mestizo | 1.00 | | | 1.00 | | |
| Native | 1.08 | 1.05–1.12 | <0.001 | 0.99 | 0.96–1.03 | 0.658 |
| Zambo, afro descendant | 1.10 | 1.06–1.14 | <0.001 | 1.02 | 0.98–1.05 | 0.388 |
| White | 1.05 | 0.99–1.14 | 0.105 | 1.01 | 0.94–1.05 | 0.769 |
| Other | 1.06 | 0.95–1.19 | 0.318 | 1.01 | 0.90–1.14 | 0.971 |
| **Economic level** | | | | | | |
| Middle | 1.00 | | | 1.00 | | |
| The poorest | 1.14 | 1.11–1.18 | <0.001 | 1.09 | 1.05–1.13 | <0.001 |
| Poor | 1.06 | 1.02–1.09 | 0.001 | 1.03 | 0.99–1.06 | 0.078 |
| Rich | 0.92 | 0.88–0.96 | <0.001 | 0.93 | 0.90–0.97 | 0.001 |
| The richest | 0.80 | 0.76–0.84 | <0.001 | 0.85 | 0.81–0.89 | <0.001 |
| **Region of origin** | | | | | | |
| Metropolitan Lima | 1.00 | | | 1.00 | | |
| Rest of the coast | 1.05 | 1.02–1.09 | 0.006 | 0.98 | 0.95–1.02 | 0.328 |
| Highlands | 1.15 | 1.12–1.19 | <0.001 | 1.00 | 0.97–1.05 | 0.709 |
| Jungle | 1.12 | 1.08–1.16 | <0.001 | 0.98 | 0.95–1.02 | 0.320 |
| **Altitude** | | | | | | |
| Less than 3 000 m.a.s.l. | 1.00 | | | 1.00 | | |
| Great or equal to 3 000 m.a.s.l. | 1.14 | 1.12–1.16 | <0.001 | 1.04 | 1.00–1.07[b] | 0.028 |
| **Health insurance** | | | | | | |
| Yes | 1.00 | | | 1.00 | | |
| No | 1.17 | 1.13–1.20 | <0.001 | 1.15 | 1.12–1.18 | <0.001 |
| **Hypertension** | | | | | | |
| Absent | 1.00 | | | 1.00 | | |
| Present | 0.71 | 0.66–0.75 | <0.001 | 0.73 | 0.68–0.78 | <0.001 |
| **Daily smoker** | | | | | | |
| No | 1.00 | | | 1.00 | | |
| Yes | 1.11 | 1.02–1.21 | 0.017 | 1.10 | 1.01–1.19 | 0.029 |

*(Continued)*

**Table 3.** (Continued)

PR = Prevalence Ratio

Prevalence Ratios are calculated with non-screening as the dependent outcome

[a] Male Sex: 95%CI (1.00063–1.051108)

[b] Altitude Greater or equal to 3000 m.a.s.l.: 95%CI (1.003953–1.072157)

## Supporting information

**S1 Data. Raw databases for Household information.**
(ZIP)

**S2 Data. Raw databases for Health information.**
(ZIP)

**S1 Table. Definition and Operationalization of Variables.**
(DOCX)

## Acknowledgments

The authors acknowledge the help from Dr. Leandro Huayanay and Dr. Armando Pezo for initial statistical guidance.

## Author contributions

**Conceptualization:** Einer Carlos Eduardo Arevalo Rios, Jennifer Layza-Reyes, Victor Hugo Noriega-Ruiz.

**Data curation:** Einer Carlos Eduardo Arevalo Rios, Jennifer Layza-Reyes.

**Formal analysis:** Einer Carlos Eduardo Arevalo Rios, Jennifer Layza-Reyes.

**Investigation:** Einer Carlos Eduardo Arevalo Rios, Jennifer Layza-Reyes, Victor Hugo Noriega-Ruiz.

**Methodology:** Einer Carlos Eduardo Arevalo Rios, Jennifer Layza-Reyes, Victor Hugo Noriega-Ruiz.

**Project administration:** Einer Carlos Eduardo Arevalo Rios.

**Resources:** Jennifer Layza-Reyes.

**Software:** Einer Carlos Eduardo Arevalo Rios, Jennifer Layza-Reyes.

**Supervision:** Victor Hugo Noriega-Ruiz.

**Validation:** Einer Carlos Eduardo Arevalo Rios, Jennifer Layza-Reyes, Victor Hugo Noriega-Ruiz.

**Visualization:** Einer Carlos Eduardo Arevalo Rios, Jennifer Layza-Reyes, Victor Hugo Noriega-Ruiz.

**Writing – original draft:** Einer Carlos Eduardo Arevalo Rios, Jennifer Layza-Reyes, Victor Hugo Noriega-Ruiz.

**Writing – review & editing:** Einer Carlos Eduardo Arevalo Rios, Jennifer Layza-Reyes, Victor Hugo Noriega-Ruiz.

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
