## [Decision Letter · Decision Letter 0]

23 May 2025

PGPH-D-25-00415

Diabetes Mellitus screening and associated factors in Peru: a cross-sectional analysis of a national health survey

Dear Dr. Arevalo Rios,

Thank you for submitting your manuscript to PLOS Global Public Health. After careful consideration, we feel that it has merit but does not fully meet PLOS Global Public Health’s publication criteria as it currently stands. Therefore, we invite you to submit a revised version of the manuscript that addresses the points raised during the review process.

Two reviewers have carefully considered your manuscript, and have made some valuable suggestions to improve the clarity of your reporting and the discussion of your work in the broader context of the literature. Please revise your manuscript and provide a point-by-point response to each reviewer's comments. 

We look forward to receiving your revised manuscript.

Kind regards,

Sarah Jose, Ph.D.

Staff Editor

Additional Editor Comments (if provided):

Reviewers' comments:

Reviewer's Responses to Questions

**Comments to the Author**

1. Does this manuscript meet PLOS Global Public Health’s publication criteria?

Reviewer #1: No

Reviewer #2: Yes

2. Has the statistical analysis been performed appropriately and rigorously?

Reviewer #1: No

Reviewer #2: Yes

3. Have the authors made all data underlying the findings in their manuscript fully available (please refer to the Data Availability Statement at the start of the manuscript PDF file)?

Reviewer #1: Yes

Reviewer #2: Yes

4. Is the manuscript presented in an intelligible fashion and written in standard English?

Reviewer #1: No

Reviewer #2: Yes

Reviewer #1: R1:

1. Good and important topic.

2. The authors need great effort to represent this topic in scientific manner.

3. The author need to:

-rearrange idea specially the objective and,

- precise selection for factors and variables of the recent study

Abstract:

4. need some details that belong to individuals who makes blood screen if:

-healthy

-apparently healthy

-have one or more than one of communicable disease

-have any health problem

-25.3% from the total number of diabetic who were screened as diabetic in Peru or according to the the research criteria?

Introduction:

5. Fine.

Methodology:

6. what is ENDES? please define all the abbreviations.

7. the researchers make questioner with self interview with participants? or from registered data of ENDES?

8. Variables: the authors need to categorize materials and methods in subtitles include:

--study design, area

- study population

- inclusion and exclusion criteria of participants.

-method of sample and data collection

-ethical consideration

-methods for data analysis

*summarize methodology by avoiding repeating of information.

*the personal and anthropometric and clinical data which obtained in the the questioner are discussed in intelligent way in methods and results.

*return to and view scientific papers.

9. Ethics: what about the questioner of the study and participants?

Results:

10. Table 1: researchers must show only the descriptive criteria of the study population(selected participants)

11. Data analysis is not meet the ADA T2DM criteria as the author mentioned for example age.

12. Table 2: -the total number of the participants in table 1 was 26166 and in table 2, 19546 participants.

13. the researcher use the age group (>=35 years) and other specific inclusion criteria, data analysis must be adjusted to the objectives of the study.

14. Table 3: this table may change as data entering change.

Discussion:

15. please discuss the results of current study and give justification of of results then compare them with previous and the more recent and similar studies,. in not more that one page.

-better to discuss directly the results only in brief, precise and scientific language.

-avoid repeating of data.

in discussion of current study the researcher open new windows which are not included in the objective of the study or calculated as variables in the data analysis.

-the authors discuss an important factor which is not included in the factors associated with the decrease screening in Peru.

-by the end of discussion you can add limitations and recommendation in points.

-acknowledgement is needed for any one who assist in this research.

Reviewer #2: Thanks for the chance to review this study on the diabetes mellitus screening and associated factors in Peru. The study uses data from the 2022 Peruvian National Demographic and Health Survey to inspect the DM screening rates and predictors in this country. The results show a quarter of adult population in Peru had DM screening. Below please find my comments and suggestions on this study.

1. Lines 100-114: considering the rich survey data used in this study, it is highly suggested that authors report the weighted prevalence of DM using the ADA criteria to address the updated knowledge on DM prevalence and thereafter enable them to calculate the proportion of undiagnosed DM. This could add significantly to the importance of this study. Also, by adding such statistics, the screening rates could be adjusted for the prevalence of DM to show the real gap on screening.

2. Results: considering the recently lowered age for DM screening and updated guidelines, the result on the statistically significant odds of screening in 18-34 age group would not infer a clinical importance.

3. Discussion: it is suggested that authors discuss the cost-effectiveness of DM screening in the broader population based on the recent ADA update and considering the shortage of resources in the context of study setting.

**Do you want your identity to be public for this peer review?** For information about this choice, including consent withdrawal, please see our Privacy Policy

Reviewer #1: **Yes: ** nahla ahmed mohammed abderhman

Reviewer #2: **Yes: ** Sina Azadnajafabad, MD, MPH

---

## [Decision Letter · Decision Letter 1]

8 Jul 2025

PGPH-D-25-00415R1

Diabetes Mellitus screening and associated factors in Peru: a cross-sectional analysis of a national health survey

Dear Dr. Arevalo Rios,

Thank you for submitting your manuscript to PLOS Global Public Health. After careful consideration, we feel that it has merit but does not fully meet PLOS Global Public Health’s publication criteria as it currently stands. Therefore, we invite you to submit a revised version of the manuscript that addresses the points raised during the review process.

Thank you for your revisions. Both reviewers responded positively to your revised manuscript; however, there are some minor details that are outstanding that should be addressed.

We look forward to receiving your revised manuscript.

Kind regards,

Daniel Parkes, PhD

Staff Editor

Journal Requirements:

Additional Editor Comments (if provided):

Reviewers' comments:

Reviewer's Responses to Questions

**Comments to the Author**

Reviewer #1: All comments have been addressed

Reviewer #2: All comments have been addressed

publication criteria?

Reviewer #1: Yes

Reviewer #2: Yes

3. Has the statistical analysis been performed appropriately and rigorously?

Reviewer #1: Yes

Reviewer #2: Yes

4. Have the authors made all data underlying the findings in their manuscript fully available (please refer to the Data Availability Statement at the start of the manuscript PDF file)?

Reviewer #1: Yes

Reviewer #2: Yes

5. Is the manuscript presented in an intelligible fashion and written in standard English?

Reviewer #1: No

Reviewer #2: Yes

Reviewer #1: General comments:

o please don't use pronouns at all (Like (we))

o define the abbreviation when used at first time, then use the symbol.

o please describe the study population to general idea of the study participants, the author many not need to account the per-centages.

o better to add variable in the description of study population.

o describe how the author applied the multivariate analysis in the current data. and avoid to write about the results.

o inclusion criteria was already mentioned. table 1: characteristics of the population.

o it is possible to find bivariate analysis using chi squire test to analyze age ?

o author can use different method for layout to avoid repartition of (ref)

o discussion need reread to rewrite in easy, intelligent, have high quality and integrity in language and ideas.

Reviewer #2: Thanks for the revision and responses. I have no further comments.

**Do you want your identity to be public for this peer review?** For information about this choice, including consent withdrawal, please see our Privacy Policy

Reviewer #1: **Yes: ** nahla ahmed mohammed abderhman

Reviewer #2: **Yes: ** Sina Azadnajafabad, MD, MPH

---

## [Decision Letter · Decision Letter 2]

7 Aug 2025

PGPH-D-25-00415R2

Diabetes Mellitus screening and associated factors in Peru: a cross-sectional analysis of a national health survey

Dear Dr. Arevalo Rios,

Thank you for submitting your manuscript to PLOS Global Public Health. After careful consideration, we feel that it has merit but does not fully meet PLOS Global Public Health’s publication criteria as it currently stands. Therefore, we invite you to submit a revised version of the manuscript that addresses the points raised during the review process.

We look forward to receiving your revised manuscript.

Kind regards,

Helen Howard

Staff Editor

Journal Requirements:

Additional Editor Comments (if provided):

Reviewers' comments:

Reviewer's Responses to Questions

**Comments to the Author**

Reviewer #1: (No Response)

Reviewer #2: All comments have been addressed

publication criteria?

Reviewer #1: Yes

Reviewer #2: Yes

3. Has the statistical analysis been performed appropriately and rigorously?

Reviewer #1: Yes

Reviewer #2: Yes

4. Have the authors made all data underlying the findings in their manuscript fully available (please refer to the Data Availability Statement at the start of the manuscript PDF file)?

Reviewer #1: Yes

Reviewer #2: Yes

5. Is the manuscript presented in an intelligible fashion and written in standard English?

Reviewer #1: No

Reviewer #2: Yes

Reviewer #1: R2:

o Minor revision is recommended.

o All previous comments of the reviewer were not addressed

o Try to avoid using pronouns (our)

o Use simple and meanfull language (Recent data from a nationally representative health survey was analyzed to find the prevalence and risk factors associated with lack diabetes screening in Peru)

o Inclusion and exclusion criteria better to be with out bulleted or numbering list.

o The personal and anthropometric and clinical data which obtained in the the questioner are discussed in intelligent way in methods, results and introduction. that mean author must not write the questioner in whole paragraph. And divided it in the paper section according to the related data.

o *please return to and view scientific papers.

Reviewer #2: The manuscript is well edited. I have no further comments.

**Do you want your identity to be public for this peer review?** For information about this choice, including consent withdrawal, please see our Privacy Policy

Reviewer #1: No

Reviewer #2: **Yes: ** Sina Azadnajafabad, MD, MPH

---

## [Editor Report · Decision Letter 3]

18 Aug 2025

Diabetes Mellitus screening and associated factors in Peru: a cross-sectional analysis of a national health survey

PGPH-D-25-00415R3

Dear Dr. Arevalo Rios,

We are pleased to inform you that your manuscript 'Diabetes Mellitus screening and associated factors in Peru: a cross-sectional analysis of a national health survey' has been provisionally accepted for publication in PLOS Global Public Health.

Best regards,

Julia Robinson

Executive Editor